# COVID-19 Vaccine-Associated Optic Neuropathy: A Systematic Review of 45 Patients

**DOI:** 10.3390/vaccines10101758

**Published:** 2022-10-20

**Authors:** Ayman G. Elnahry, Mutaz Y. Al-Nawaflh, Aisha A. Gamal Eldin, Omar Solyman, Ahmed B. Sallam, Paul H. Phillips, Abdelrahman M. Elhusseiny

**Affiliations:** 1Department of Ophthalmology, Faculty of Medicine, Cairo University, Cairo 11956, Egypt; 2Division of Epidemiology and Clinical Applications, National Eye Institute, National Institutes of Health, Bethesda, MD 20892, USA; 3Division of Ophthalmology, King Hussein Hospital, Jordanian Royal Medical Services, Amman 11855, Jordan; 4Maryland Eye Care Center, Silver Spring, MD 20910, USA; 5Department of Ophthalmology, Research Institute of Ophthalmology, Giza 11261, Egypt; 6Department of Ophthalmology, Qassim University Medical City, Al-Qassim 52571, Saudi Arabia; 7Department of Ophthalmology, Harvey and Bernice Jones Eye Institute, University of Arkansas for Medical Sciences, Little Rock, AR 72205, USA

**Keywords:** CNS inflammation, COVID-19, ischemic optic neuropathy, ocular inflammation, optic neuritis, optic neuropathy, vaccination

## Abstract

We provide a systematic review of published cases of optic neuropathy following COVID-19 vaccination. We used Ovid MEDLINE, PubMed, and Google Scholar. Search terms included: “COVID-19 vaccination”, “optic neuropathy”, “optic neuritis”, and “ischemic optic neuropathy”. The titles and abstracts were screened, then the full texts were reviewed. Sixty eyes from forty-five patients (28 females) were included. Eighteen eyes from fourteen patients (31.1%) were diagnosed with anterior ischemic optic neuropathy (AION), while 34 eyes from 26 patients (57.8%) were diagnosed with optic neuritis (ON). Other conditions included autoimmune optic neuropathy and Leber hereditary optic neuropathy. Fifteen patients (33.3%) had bilateral involvement. The mean age of all patients was 47.4 ± 17.1 years. The mean age of AION patients was 62.9 ± 12.2 years and of ON patients was 39.7 ± 12.8 years (*p* < 0.001). The mean time from vaccination to ophthalmic symptoms was 9.6 ± 8.7 days. The mean presenting visual acuity (VA) was logMAR 0.990 ± 0.924. For 41 eyes with available follow-up, the mean presenting VA was logMAR 0.842 ± 0.885, which improved to logMAR 0.523 ± 0.860 at final follow-up (*p* < 0.001). COVID-19 vaccination may be associated with different forms of optic neuropathy. Patients diagnosed with ON were more likely to be younger and to experience visual improvement. More studies are needed to further characterize optic neuropathies associated with COVID-19 vaccination.

## 1. Introduction

In 2021, vaccination against severe acute respiratory syndrome coronavirus 2 (SARS-CoV-2) became a primary focus of public health efforts to control the COVID-19 pandemic. Even though they were generally found to be safe and effective in multiple large controlled clinical trials, the relatively fast and wide deployment of COVID-19 vaccines has made them a subject of considerable scrutiny and analysis since the time of their introduction to the public.

The COVID-19 disease itself has affected the eye in many ways. Previous research demonstrated a link between COVID-19 infection and ophthalmic manifestations, both directly and indirectly. For example, it was reported that inflammatory conditions such as conjunctivitis, scleritis, orbital inflammation, keratitis, and retinal affection may be directly linked to COVID-19 infection [1,2,3,4,5,6,7,8,9]. Regarding indirect impact, several studies have addressed the relationship between eye strain and dry eye symptoms and the increased screen time in both pediatric and adult populations during the pandemic [7,10].

Since their deployment, COVID-19 vaccines have been linked to various ophthalmic manifestations [11,12]. These manifestations have involved the orbit [13], cornea [14,15], uvea [16,17], optic nerve [18,19], retina [20,21], and retinal vessels [22]. Regarding neuro-ophthalmic manifestations, several cranial nerve palsies were reported following COVID-19 vaccines that involved the oculomotor [23], abducens [24], and facial nerves [25,26,27,28]. Recently, multiple reports were published on the development of optic neuropathy following COVID-19 vaccination [19,29,30,31,32,33,34]. In this study, we provide a systematic review of all cases of optic neuropathy following COVID-19 vaccination published to date.

## 2. Materials and Methods

We performed a systematic literature search using Ovid MEDLINE, PubMed, and Google Scholar for published cases of optic neuropathy that followed COVID-19 vaccination up to 10 September 2022. We used a combination of the following terms: “COVID-19 vaccination”, “optic neuropathy”, “optic neuritis”, “papillitis”, “retrobulbar optic neuritis”, “ischemic optic neuropathy”, “NAION”, and “AION”. We initially screened titles and abstracts for the identification of studies, then the full texts were retrieved for eligible studies for a complete review and inclusion in the final analysis. We only included cases that were published in the English language, peer-reviewed, and that included details on optic nerve involvement. There were no restrictions on study type, and all studies, including case reports and case series, were eligible for inclusion. Exclusion criteria included insufficient evidence or details of optic nerve involvement.

### Data Extraction and Statistical Analysis

The following data were extracted from the included studies: type of study, age of patients, gender of patients, type of vaccine, the dose of vaccination, the duration between administration of the vaccine and the onset of ocular symptoms, the presenting, and the final visual acuity, presenting symptoms and signs, results of systemic and ocular investigations, diagnoses, and mode of treatment. For continuous variables, we reported the mean as mean ± standard deviation. We made comparisons of populations, when appropriate, using a two-tailed two-sample t-test for means. A *p*-value of <0.05 was considered statistically significant.

## 3. Results

### 3.1. Studies and Patients

We identified 29 studies (21 case reports and 8 case series) that reported on patients that developed optic neuropathy following COVID-19 vaccination. From those, a total of 60 eyes from 45 patients that developed optic neuropathy following COVID-19 vaccination met the inclusion criteria (Table 1). Eighteen eyes from fourteen patients (31.1%) were diagnosed with anterior ischemic optic neuropathy (AION), while 34 eyes from 26 patients (57.8%) were diagnosed with optic neuritis. One patient was diagnosed with papillitis in one eye and neuroretinitis in the other, while four eyes from three patients were diagnosed with autoimmune optic neuropathy (AON). In addition, one patient was diagnosed with Leber hereditary optic neuropathy (LHON), which was genetically confirmed (m.14568A > G mutation in MT-ND6). Fifteen patients (33.3% of the total), therefore, had bilateral optic nerve involvement. Regarding AION, five eyes from three patients were diagnosed with arteritic anterior ischemic optic neuropathy (AAION), all confirmed by temporal artery biopsy, while the remaining eyes (13 eyes from 11 patients) were diagnosed as non-arteritic anterior ischemic optic neuropathy (NAION). In addition to optic neuropathy, one patient was diagnosed with central nervous system (CNS) inflammatory syndrome, one with acute disseminated encephalomyelitis (ADEM), three with giant cell arteritis (GCA), three with neuromyelitis optica spectrum disorder (NMOSD), and five with myelin-oligodendrocyte-glycoprotein (MOG) antibody-associated disease.

### 3.2. Demographic Data

Twenty-eight patients (62.2%) were females, while 17 (37.8%) were males. The mean age of all patients was 47.4 ± 17.1 years (Range: 15–87 years). The mean age of AION patients was 62.9 ± 12.2 years (Range: 40–87 years), while the mean age of optic neuritis patients was 39.7 ± 12.8 years (Range: 19–65 years) (*p* < 0.001). The mean time from vaccination to onset of ophthalmic symptoms was 9.6 ± 8.7 days (Range: 0–42 days).

### 3.3. Visual Acuity Outcomes

The mean presenting visual acuity of all patients was logMAR 0.990 ± 0.924 (20/200 in Snellen notation). Of the 41 eyes (32 patients) that had both presenting and final follow-up visual acuity, 21 eyes (52.5%) from 18 patients (56.3%) experienced an improvement in visual acuity, with 16 eyes (76.2%) due to optic neuritis, 3 (14.3%) due to AION, and 2 (9.5%) due to AON. For those 41 eyes, the mean presenting visual acuity was logMAR 0.842 ± 0.885 (20/140 in Snellen notation), while at final follow-up, it significantly improved to logMAR 0.523 ± 0.860 (20/70 in Snellen notation; *p* < 0.001). For the 18 eyes (14 patients) with AION, the mean presenting visual acuity was logMAR 0.987 ± 0.929 (20/200 in Snellen notation), comparable to the 32 eyes (25 patients) with optic neuritis (2 eyes from one patient had no reported visual acuity), who had mean presenting visual acuity of logMAR 1.068 ± 0.980 (20/230 in Snellen notation) (*p* = 0.772). However, for cases with available follow-up visual acuity, the final visual acuity for 14 eyes (11 patients) with AION was logMAR 0.970 ± 1.012 (20/185 in Snellen notation), which was significantly worse than that for 21 eyes (17 patients) with optic neuritis with available follow-up (final visual acuity of logMAR 0.227 ± 0.659 (20/35 in Snellen notation), *p* = 0.025).

**Table 1 vaccines-10-01758-t001:** Summary of cases of optic neuropathy following COVID-19 vaccination.

Author	Article Type	Age	Sex	Vaccine, Dose	Time from Vaccine to Symptoms (Days)	Baseline VA	Eye	Systemic Conditions (Pre-Vaccination)	Manifestations	Outcome	Final VA
Elnahry et al., 2021 [19]	CS	69	F	BNT162b2, #2	16	CF	OD	Hypertension, DM, cutaneous T-cell lymphoma	Blurry vision OU with immediate OS clearing but persistent blurring OD. The exam showed optic nerve head edema (OD > OS) and RAPD OD. OCT imaging and CSF confirmed a diagnosis of CNS inflammatory syndrome with neuroretinitis OD and papillitis OS.	Significant improvement of optic disc edema with stable vision after 5 days of IV methylprednisolone.	CF
						20/20	OS				20/20
		32	F	COVISHIELD, #1	4	20/30	OS	None	Blurred vision with superior field defect OS. Examination revealed optic disc swelling and RAPD with decreased RNFL thickness. MRI was consistent with optic neuritis	Significant improvement of optic nerve head swelling and improved VF defect and VA after 3 days of IV methylprednisolone followed by PO prednisone.	20/20
Garcia-Estrada et al., 2021 [29]	CR	19	F	Ad26.COV2.S, #1	7	20/20	OS	None	Ocular pain and vision loss OS, with exam revealing amaurosis, RAPD, and papillitis.	Resolution of symptoms and papillitis after 5 days of IV methylprednisolone followed by a PO prednisolone.	20/20
Girbardt et al., 2021 [30]	CR	67	M	Vaxveria, #1	2	20/200	OD	DM, hypercholesterolemia	Decreased vision and scotomas OD with exam revealing an elevated, congested optic nerve head with surrounding intraretinal hemorrhages and cotton-wool spots.NAION was diagnosed	NR	NR
Leber et al., 2021 [31]	CR	32	F	Corona Vac, #2	0	20/200	OS	NR	Rapidly progressive worsening vision and pain with EOM OS. Exam revealed RAPD OS and disc swelling OD and OS. MRI revealed optic neuritis OU.	Improvement in symptoms and vision after 5 days of IV methylprednisolone.	20/20
						20/20	OD				20/25
Nachbor et al., 2021 [32]	CR	64	F	BNT162b2, #1	6	20/80	OS	DM	Acute, painless, unilateral vision loss with superior sectoral optic disc edema OS after 1st dose. After 2nd dose, VA was CF with persistent APD OS. RNFL OCT showed diffuse thickening OS. NAION was diagnosed OS.	Improvement of symptoms with PO prednisone over 1 week. Resolution of optic disc edema followed by optic nerve pallor.	20/100
Pawar et al., 2021 [33]	CR	28	F	AstraZeneca, #1	21	20/120	OS	None	Sudden vision loss OS, with exam revealing mild blurring of the optic disc margin. MRI was consistent with optic neuritis OS.	Resolution of symptoms after IV methylprednisolone followed by PO steroid.	20/20
Maleki et al., 2021 [34]	CR	79	F	BNT162b2, #2	2	20/1250	OD	None	Bilateral sudden loss of vision, OD>OS, with 3+ RAPD OD. OCT, FA, ICG, and temporal artery biopsy consistent with consistent with bilateral AAION.	Initiated on subcutaneous tocilizumab. Prognosis was NR.	NR
						20/40	OS				NR
Tsukii et al., 2021 [35]	CR	55	F	BNT162b2, #1	3	20/20	OD	None	Visual disturbance with RAPD OD. Fundoscopy revealed diffuse optic disc swelling OD. An inferior VF defect suggesting AION.	vision remained normal and there was diffuse pallor OD, although no treatment was initiated	20/20
Lin et al., 2022 [36]	CR	61	F	ChAdOx1nCoV-19 #1	7	20/50	OS	Hypertension, hyperlipidemia	Scotoma in inferior field with hazy vision OS and headache. Fundus exam: optic disc edema OS. VF: inferior altitudinal field defect. OCT: peripapillary RNFL edema and GCL thinning in the superior macula. FA: filling delay, decreased choroidal perfusion, and optic disc leakage consistent with NAION OS.	PO prednisolone with gradual tapering. After 6 weeks, VA became 20/80 OS, and disc edema resolved	20/80
Chung et al., 2022 [37]	CR	65	F	AstraZeneca #2	15	CF	OD	None	RAPD OD. Fundus exam: a swollen disc with several splinter hemorrhages OD. VF: inferior arcuate and cecocentral visual field defects. OCT: thickened peripapillary RNFL and thinning of RNFL. MRI: no increased signal intensity or abnormal enhancement. NAION OD diagnosed	IV methylprednisolone followed by PO steroid taper. VA improved to 20/200 with optic disc pallor and no improvement in VF defect.	20/200
Sanjay et al., 2022 [38]	CR	52	F	COVISHIELD #2	4	20/20	OS	DM	Blurring of temporal disc margin with hyperemia OS. RAPD OS. Color vision was abnormal OU. NAION OS diagnosed.	PO aspirin for 1 month. BCVA OS was 20/20 at 1 month with resolved disc edema.	20/20
Roy et al., 2022 [39]	CS	27	F	COVISHIELD #1	4	20/200	OS	None	Progressive blurring of vision OS. RAPD and color desaturation OS. Fundus exam: diffuse swelling of the optic nerve head. VF: an enlarged blind spot. MRI brain and orbit: enhancement of left optic nerve just behind the disc. VEP: flat wave OS compared to OD. Diagnosed as optic neuritis OS.	IV methylprednisolone followed by PO steroid taper. BCVA improved to 20/40 OS with decreased disc swelling.	20/40
		48	F	COVISHIELD #2	2	20/80	OS	NR	RAPD OS. Fundus: swollen optic disc with blurred margins. OCT: peripapillary swelling of the retina. VF: an inferior arcuate defect. VEP: delayed latency and decreased amplitude OS. Optic neuritis OS was diagnosed	IV methylprednisolone. BCVA improved to 20/30 OS with improved VF.	20/30
		40	M	COVISHIELD #1	5	20/200	OD	NR	Sluggishly reacting pupils OU. Fundus: bilaterally blurred and swollen optic disc margin. VF: generalized depression OU. VEP: flat waves. Diagnosis of bilateral optic neuritis was made.	BCVA and VF improved OU after steroid therapy.	20/30
						20/200	OS				20/40
Elhusseiny et al., 2022 [40]	CR	51	M	BNT162b2 #2	1	CF 3 ft	OS	DM	Fundus exam: optic disc edema, peripapillary hemorrhages, and blunted foveal reflex OS. FA: optic disc leakage OS. OCT: marked thickening of the peripapillary retina, intraretinal fluid and hyperreflective foci consistent with exudates, and subretinal fluid under the fovea. NAION OS diagnosed.	PO prednisone over 1 month. Disc swelling and subretinal fluid resolved with BCVA of 20/400 OS.	20/400
Madina et al., 2022 [41]	CS	65	F	BNT162b2 #1	5	PL	OD	Medullary thyroid cancer, hypothyroidism, prediabetic, hyperlipidemia	Had vision loss and pain with eye movements OD. RAPD OD, optic disc swelling associated with cotton-wool spots and flame hemorrhages. MRI orbits: evidence of right optic neuritis. Optic neuritis OD diagnosed.	IV methylprednisolone and IVIG BCVA improved to 20/100 OD.	20/100
		67	M	Moderna #2	1	20/40	OD	Prediabetic, hyperlipidemia	bilateral eye redness, chemsosis, and blurring of vision. Had RAPD OS. MRI orbits: evidence of left optic neuritis. Optic neuritis OU diagnosed.	IV methylprednisolone. BCVA improved to 20/20 OD with a normal visual field, but he continued to have NPL OS.	20/20
						NPL	OS				NPL
Franco et al., 2022 [42]	CS	53	M	BNT162b2/Comirnaty vaccine #2	10	20/20	OD	None	RAPD OS and fundus exam showed optic disc swelling with peripapillary hemorrhages OU. Disc edema confirmed by OCT OU. VF: constriction of peripheral visual field OS and an incomplete lower nasal scotoma OD. NAION OU diagnosed.	Stable BCVA with sluggish pupils. Fundus exam: pale discs OU without any hemorrhages. VF did not change OD, but OS was slightly better.	20/20
						20/40	OS				20/40
		65	M	BNT162b2/Comirnaty vaccine #1	12	20/200	OD	Hypertension	RAPD with optic disc swelling and peripapillary hemorrhages OD. NAION diagnosed.	No specific treatment was given. No vision improvement with dyschromatopsia (Ishihara test: 1/17), temporal optic disc pallor, and blind spot enlargement with centrocecal scotoma on VF.	20/200
Norman et al., 2022 [43]	CS	62	M	BNT162b2 #1	6	20/20	OS	None	RAPD OS. Inferior optic nerve swelling OS. VF: superior altitudinal defect. Diagnosed as presumed AON OS.	Stable BCVA and color vision, while VF continued to show superior altitudinal defect. Had complete resolution of disc edema, with some pallor and loss of the RNFL inferiorly.	20/20
		48	F	BNT162b2 #1	5	20/70	OD	Hypertension, migraine, asthma, trigeminal neuralgia	Trace RAPD OS. VF: paracentral scotoma OD with full field and no scotomas OS. Optic nerve head: temporal pallor and trace edema OD and temporal pallor OS. Labs: elevated ESR and CRP. Diagnosed as papillitis OU due to AON.	Started on steroids.VA improved OU, RAPD disappeared, with resolved disc swelling.	20/50
						20/200	OS				20/80
		38	M	BNT162b2 #1	3	20/25	OS	Hypertension, hyperlipidemia	Color Plates showed 12/14 OS with 2+ RAPD. VF: superior and inferior arcuate defects OS. Optic nerve head had a 0.1 cup-to-disc ratio with severe pallid edema, enlarged capillaries, and small disc hemorrhages. Diagnosed as presumed AON OS	IV methylprednisolone followed by PO steroid taper. Stable VA, persistent 2+ RAPD OS, with improved disc swelling and VF defects.	20/25
Rizk et al., 2022 [44]	CR	15	M	BNT162b2 #2	7	20/200	OD	None	No RAPD. Color vision: 8/17 OD and 6/17 OS. Hyperemic telangiectatic vessels on optic disc OU. VF: cecocentral scotoma OU. Diagnosis of LHON OU was confirmed genetically.	Started on idebenone 300 mg 3 times daily and counselled about lifestyle changes and triggers.	NR
						20/200	OS				NR
Kumar et al., 2022 [45]	CR	73	M	COVISHEILD #1	5	20/1200	OD	None	Bilateral sluggish and poorly sustained pupillary reactions. Exam: edematous disks and chorioretinal changes inferonasal to the disk OS. NAION diagnosed OU	Patient did not report any appreciable visual gain.	20/1200
						20/120	OS				20/120
Che et al., 2022 [46]	CR	87	F	BNT162b2 #1	1	HM	OD	Hypertension	Bilateral optic disc edema with focal disc hemorrhage. FFA: peripapillary choroidal filling delay in the vertical watershed zone OD. MRI: circumferential enhancement of the intraorbital portion of the optic nerve sheath bilaterally. Biopsy of the right temporal artery confirmed the diagnosis of GCA and AAION OU.	IV methylprednisolone followed by PO steroid taper. BCVA improved to 0.1 logMAR OS but worsened to NPL OD at 4 months after treatment.	NPL
					4	20/30	OS				20/25
Raxwal et al., 2022 [47]	CR	47	F	Moderna #1	8	20/50	OS	Hypertension	Blurring and RAPD OS. There was no pallor of the optic nerve and no evidence of papilledema OU. Optic neuritis diagnosed OS	IV methylprednisolone followed by PO steroid taper.. Follow-up revealed normal eye exam and normal VA.	20/20
Wang et al., 2022 [48]	CS	21	F	Sinopharm #1 and #2	42 and 21	20/30	OD	None	RAPD OD. Fundus exam: blurred optic disc margin with congestion and edema OD. FA: early hyperfluorescence and late enhancement of the right optic papilla. OCT: significant thickening of the RNFL. VF: central scotoma. VEP: decreased amplitude. MRI brain: a small ischemic focus in the left frontal lobe. orbital MRI: no significant abnormalities. Optic neuritis diagnosed OD.	IV methylprednisolone followed by PO steroid taper. Papillary congestion and edema OD gradually resolved. BCVA recovered to 1.0 after 1 month.	20/20
		38	F	Sinopharm #1	21	CF 1 m	OD	None	RAPD OD. Fundus exam: blurred borders of the optic disc with congestion and edema OD. FA: early hyperfluorescence of the optic papilla OD with late staining. OCT: significant thickening of RNFL. VEP: prolonged P100 wave latency and decreased amplitude. VF: centripetal narrowing. Orbital CT: hypointense thickening of optic nerve. Optic neuritis diagnosed OD.	IV methylprednisolone followed by PO steroid taper. Papillary congestion and edema gradually resolved OD. BCVA recovered to 1.0 after 1 month.	20/20
Haseeb et al., 2022 [49]	CR	40	M	BNT162b2 #1	4	20/40	OS	DM	Vision loss OS. Exam: RAPD OS and an edematous pale optic disc with blurred edges and splinter hemorrhages. FA: early leakage OS with late staining. NAION diagnosed OS	NR	NR
Helmchen et al., 2022 [50]	CR	40	F	AstraZeneca #1	14	NR	OD	Multiple sclerosis	Progressive diminution of vision over 48 h. CSF: severe pleocytosis, increased lactate and strongly elevated protein. Cranial MRI: numerous old white matter lesions compatible with MS and increased signal intensity in the chiasm, optic nerves and tracts. Mild optic chiasm enhancement was observed. VEP: unrecordable OU. Spinal MRI: increased longitudinal centrally located signal intensities throughout the thoracic myelon. Diagnosed with NMOSD and optic neuritis OU.	Two days after receiving IV methylprednisolone, there was no contrast enhancement visible. she was also treated with plasmapheresis and immunoadsorption with slight recovery of visual functions but paraplegia, loss of sensory function below T5, and incontinence persisted. Two months after subacute onset, with even more improved BCVA but unchanged paraplegia follow-up spinal	NR
						NR	OS				NR
Nagaratnam et al., 2022 [51]	CR	36	F	ChAdOx1nCoV-19 #1	12	20/50	OD	None	CSF analysis on day 2 of admission showed a normal protein with pleocytosis. CSF oligoclonal IgG bands were present, suggestive of intrathecal IgG synthesis. VEP: unrecordable OS and delayed OD, consistent with demyelinating pathology of anterior visual pathways OU but OS>OD. MRI brain: multiple T2/ FLAIR hyperintense lesions in subcortical white matter, posterior limb of bilateral internal capsules, pons and left middle cerebellar peduncle. No definite abnormal signal or enhancement of optic nerves. Patient diagnosed with ADEM and optic neuritis OU.	IV methylprednisolone followed by PO steroid taper. VA improved to near baseline with full color vision OU. Both optic nerves were pale.	20/16
						20/100	OS				20/20
Shirah et al., 2022 [52]	CR	31	F	BNT162b2 #1	14	20/20	OS	Systemic lupus erythematosus	Diagnosed with SLE 10 years earlier. Fundus exam: normal OU without optic disc swelling. VF: paracentral VF contraction OU. Aquaporin-4 IgG antibody titer was positive at 1:1000. OCT: mild paracentral optic nerve thickening OS. VEP: mild to moderate prechiasmatic optic pathway dysfunction OS with secondary axonal loss. MRI: abnormal signal intensity and enhancement within the intraocular and intraorbital optic nerve OS. Optic neuritis diagnosed OS.	IV methylprednisolone followed by plasmapheresis, however, there was: no improvement. Rituximab was also started. Blurred vision OS remained unchanged, but ocular pain subsided.	20/20
Pirani et al., 2022 [53]	CS	31	F	BNT162b2 #1	6	20/200	OD	Ankylosing spondylitis	Fundus exam): mildly blurred margins of optic disc OD. VF showed scotoma OD. MRI brain, orbits and spine: no demyelination. T1-weighted MRI brain and orbits: enhancement of retrobulbar optic nerve OD; diagnosed with retrobulbar optic neuritis OD.	IV methylprednisolone followed by PO prednisone taper. On the third day BCVA improved to 20/20 OD.	20/20
		46	F	BNT162b2 #1	8	20/40	OD	Hashimoto thyroiditis	Slit-lamp exam OU was unremarkable. VF testing and MRI confirmed the diagnosis of retrobulbar optic neuritis OD.	IV methylprednisolone followed by PO steroid taper. BCVA improved to 20/25 OD and VF deficit resolved.	20/25
Singu et al., 2022 [54]	CR	39	F	BNT162b2 #1	12	20/20	OS	None	Ocular pain OS and headache without any other neurologic deficits. MRI: slight left optic neural swelling and perineuritis OS. Anti- MOG antibody: positive Diagnosed with optic neuritis and perineuritis OS.	Visual disturbance never recurred, and her ocular pain and headache subsided only with anti-inflammatory agents.	20/20
Xia et al., 2022 [55]	CR	68	M	ChAdOx1nCoV-19 #2	29	NPL	OS	Chronic obstructive pulmonary disease	Bilateral jaw claudication and profound lethargy, but no scalp tenderness, fever, weight loss, and no shoulder, neck, or hip pain. Labs: normal ESR with mildly elevated CRP. Bilateral temporal artery biopsy confirmed GCA. AAION diagnosed OS.	Treated with oral and IV steroids. Had episodes of blurred vision OD on day 3 so a fourth dose of methylprednisolone was given.	NPL
Netravathi et al., 2022 [56]	CS	29	F	ChAdOx1nCoV-19 #1	11	HM	OD	NR	RAPD OD. Anti-MOG- positive VEP: absent waveform OD, normal OS. MRI brain: T2 /FLAIR hyperintensity of long intraorbital segment of optic nerve OD with contrast enhancement. MOG-antibody-associated optic neuritis diagnosed OD.	IV methylprednisolone followed by PO steroid taper and plasmapheresis.	NR
		39	M	ChAdOx1nCoV-19 #1	14	CF	OD	NR	RAPD OD. VF: right inferonasal quadrant involvement. MOG-antibody-associated optic neuritis OD diagnosed.	IV methylprednisolone followed by PO steroid taper	NR
		54	M	ChAdOx1nCoV-19 #1	14	20/40	OS	NR	RAPD OS. VEP: normal OD, absent waveform OS. Anti-MOG: positive. MRI brain and spine: hyperintensity in Rt pons. MOG-associated optic neuritis diagnosed OS.	IV methylprednisolone followed by PO steroid taper	NR
		34	M	ChAdOx1nCoV-19 #1	1	PL	OD	NR	Non-reactive pupil OD. VEP: absent waveform OD. MRI: optic nerve tortuosity with prominent perioptic sheath and fat stranding OD. Optic neuritis diagnosed OD.	IV methylprednisolone followed by PO steroid taper	NR
		45	F	ChAdOx1nCoV-19 #1	21	20/40	OD	NR	RAPD OS. Normal pupillary reaction OD. MOG-associated optic neuritis diagnosed OU.	IV methylprednisolone followed by PO steroid taper and plasmapheresis.	NR
						HM	OS				NR
		30	M	ChAdOx1nCoV-19 #1	14	NPL	OD	NR	Optic disc edema OU. VEP: non-recordable OU. CSF lymphocytosis. MRI brain: subcortical hyperintense foci in both cerebral hemispheres. MRI Optic nerves: OD>OS intraneural hyperintensities Optic neuritis diagnosed OU.	IV methylprednisolone followed by plasmapheresis and Rituximab.	NR
						20/600	OS	NR			NR
		40	M	ChAdOx1nCoV-19 #1	10	20/60	OD	NR	Serum MOG positive. MRI brain: T2 Hyperintensities in pons, bilateral thalami, right frontal cortex. MRI spine: longitudinally extensive myelitis. MOG-associated opticomyelopathy with optic neuritis OU diagnosed.	IV methylprednisolone followed by PO steroid taper and mycophenolate mofetil.	NR
						20/60	OS	NR			NR
		65	F	ChAdOx1nCoV-19 #1	42	HM	OD	NR	VEP OD non-recordable. CSF cells. Elevated ESR. NMO antibodies: positive. MRI brain: few hyperintensities in frontal subcortex. MRI Spine: hyperintensity with patchy contrast enhancement and bright spotty areas. NMO with optic neuritis OD diagnosed.	Received 3 cycles plasmapheresis followed by IV methylprednisolone then PO prednisolone 40 mg PO and mycophenolate mofetil.	NR

Abbreviations. #1 indicates that the event occurred following the first dose of vaccination, while #2 indicates that it occurred following the second dose. AAION = arteritic anterior ischemic optic neuropathy, AION = anterior ischemic optic neuropathy, ADEM = acute disseminated encephalomyelitis, ANA = anti-nuclear antibodies, AON = autoimmune optic neuropathy, BCVA = best corrected visual acuity, CF = counting fingers, CR = case report, CS = case series, CNS = central nervous system, CSF = cerebrospinal fluid, GCA = giant cell arteritis, DM = diabetes mellitus, ESR = erythrocyte sedimentation rate, GCL = ganglion cell layer, FA = fluorescein angiography, HM = hand movement, ICG = indocyanine green, IV = intravenous, LHON= Leber hereditary optic neuropathy, MOG = myelin-oligodendrocyte-glycoprotein, MRI = magnetic resonance imaging, MS = multiple sclerosis, NAION = non-arteritic anterior ischemic optic neuropathy, NMOSD = neuromyelitis optica spectrum disorder, NPL = no perception of light, NR = not reported, OCT = ocular coherence tomography, OD = right eye, OS = left eye, OU = both eyes, PL = perception of light, PO = oral, RAPD = relative afferent pupillary defect, RNFL = retinal nerve fiber layer, SLE = systemic lupus erythematosus, VA = visual acuity, VDRL = venereal disease research laboratory, VEP = visual evoked potential, VF = visual field.

## 4. Discussion

In this systematic review of published cases of optic neuropathy following COVID-19 vaccination, we found that COVID-19 vaccination was associated with several forms of optic neuropathy, most commonly AION and optic neuritis. All subtypes of COVID-19 vaccines, including mRNA, viral vector, and inactivated viral vaccines were associated with optic neuropathy. However, protein subunit vaccines, such as the Novavax vaccine, were not reported as a cause of optic neuropathy in the current review. The temporal association between vaccine administration and the development of optic neuropathies in these cases makes a causal link plausible, with a mean time from vaccination to the development of ocular symptoms of 9.6 ± 8.7 days. Cases with a late onset of optic neuropathy, however, are less likely to be related to vaccination and could be coincidental. Vaccines and their adjuvants are meant to robustly activate the innate immune system, and adaptive immunity then follows. Overactivation of this response, however, may occur in some patients and lead to rare immune-mediated complications.

AION is an important cause of loss of vision in adults and is classically divided into AAION and NAION. Previously, the incidences per 100,000 individuals for NAION and AAION, respectively, were reported as 2.30 and 0.36 [57]. Regarding pathogenesis, AAION results from inflammation and thrombosis of the short posterior ciliary arteries, which causes optic nerve head infarction [58,59]. It occurs mainly in the setting of GCA. It is an ophthalmic emergency and requires immediate treatment with systemic steroids [58]. NAION is classically idiopathic [58], though there is an association with various conditions including sleep apnea [60,61,62], certain drugs such as sildenafil and interferon [63,64,65], and ocular conditions such as optic disc drusen and crowded discs [66,67,68]. It commonly manifests as an altitudinal field defect. Post-vaccination NAION was also reported following the administration of the influenza vaccine [69,70].

Most of the post-vaccination inflammatory syndromes affecting the CNS were related to influenza vaccines, and optic neuritis is the most common clinical presentation of these syndromes [71]. The number of people receiving COVID-19 vaccines annually is even larger than those who receive influenza vaccines. This possibly makes the appearance of complications following COVID-19 vaccination seem more common. Certain individuals may also be at a higher risk of developing complications such as patients with diabetes, hypertension, or patients with autoimmune diseases, as suggested in the current review. The development of cerebral venous sinus thrombosis in women under 50 years of age was associated with the AstraZeneca-Oxford COVID-19 vaccine owing to a vaccine-induced immune thrombotic thrombocytopenia [72]. This complication was excluded by magnetic resonance venography in most of the reported cases of post-COVID-19 vaccination optic neuropathy included in this review.

Historically, gender type seems to influence the incidence and type of optic neuropathy. Previous studies have demonstrated a female predominance of incident optic neuritis, with one large study from the UK demonstrating that almost 70% of new cases over 22 years involved females [73]. Lee et al., however, previously showed in a cohort of patients with diabetes that the male gender increases the risk of developing AION by around 32% [74]. In the current review, we found that 62.2% of cases of optic neuropathy following COVID-19 vaccination occurred in females. This percentage was even higher (73.1%) when looking at only patients who developed optic neuritis, indicating a possible higher risk of optic neuropathy, especially optic neuritis, in females following COVID-19 vaccination. This could be due to hormonal or genetic differences and requires further analysis in larger prospective studies.

Ocular side effects including ocular inflammation, were reported following COVID-19 vaccination but are thought to be rare. In a relatively large multinational study of ocular inflammatory events following COVID-19 vaccination, 70 patients were reported to develop ocular inflammation within 14 days of COVID-19 vaccination, but only 2 (2.9%) patients were diagnosed with optic neuritis [75]. This indicates that the incidence of optic neuropathy following COVID-19 vaccination could be low. The latter study, however, did not provide specific details on cases with optic nerve involvement and so were not included in this review.

We have previously described that an autoimmune mechanism underlies the development of optic neuropathies following vaccination [11]. Previously, Stübgen et al. reported that there is no long-term risk of developing optic neuropathy following vaccination, but that the presence of adjuvants contributes to the process [76]. However, in the absence of adjuvants in several of the COVID-19 vaccines, this explanation is insufficient [19]. Clinically, it is challenging to differentiate between AION and optic neuritis, and the diagnosis is usually based on both clinical impression and multimodal imaging findings including neuroimaging [77,78,79]. Some differentiating features include older age of onset, altitudinal field defect, and worse visual outcomes in patients with AION, which was also found in our study; however, these features cannot confidently distinguish between both conditions [77]. Furthermore, the pathophysiology (and treatment) of both types of optic neuropathies is suggested to be different and it is not currently clear why some patients develop AION while others develop optic neuritis following vaccination. Documented risk factors for NAION include small cup-to-disc ratio, diabetes, hyperlipidemia, and hypertension, and it is likely that the development of NAION is a multifactorial process that includes the pre-existing structural compromise of the optic nerve [36,80,81].

Tsukii et al. have proposed that neutralizing antibodies directed against SARS-CoV-2 spike proteins after vaccination may cross-react with proteins in the retinal vasculature and retinal pigment epithelial cells, a mechanism also endorsed by Maleki et al. [34,35]. It is possible that these antibodies also cross-react with elements of the CNS including the optic nerve, a phenomenon suggested by the co-occurrence of CNS disease in several patients reported in this review. It is, therefore, conceivable that both processes may have played a role in the development of optic neuropathy in the cases reported herein. This is also supported by reports on cases of optic neuritis that developed following COVID-19 infection and was suggested to be due to molecular mimicry between viral antigens, which are also partly present in the vaccines, and CNS proteins [82,83,84]. Another possible explanation for the development of post-vaccination optic neuropathy that could link inflammation and ischemia was recently elucidated by Francis and colleagues in patients that developed optic neuropathy with immune checkpoint inhibitors used for the treatment of cancer [85]. These immunotherapy agents, such as vaccines, enhance the adaptive immune response resulting in a range of adverse inflammatory events including both ophthalmic and neurologic phenomena [85]. Francis and colleagues also indicated that this class of drugs could result in optic papillitis, a specific type of optic neuritis that involves the optic nerve head, leading to ischemia of the optic nerve head and an AION-like picture [85,86].

This review has limitations, including that it relies on a cohort of a relatively small number of case reports. Larger case-control studies would have provided a more optimal analysis of the association and the temporal relationship between COVID-19 vaccines and optic nerve disease, but unfortunately, no studies exist to date. Optic neuritis and AION are among the most common optic neuropathies even among the unvaccinated, with predisposing risk factors similar to those seen in patients in the current report. Therefore, the association of optic neuropathy with vaccination in many patients could be coincidental and unrelated to vaccination. Furthermore, the number of cases of optic neuropathy reported to date is very small, despite billions of individuals having already received COVID-19 vaccines, suggesting that the incidence of this complication is very low. The frequency of optic neuropathy following COVID-19 vaccination, however, is currently unknown and cannot be determined based on the available data.

## 5. Conclusions

In conclusion, several cases of optic neuropathy were reported following the administration of COVID-19 vaccines, suggesting an association and perhaps a cause–effect relationship, with at least one case reporting a positive rechallenge phenomenon following the second dose of vaccination [48]. Nevertheless, the benefits of vaccination against SARS-CoV-2 are substantial and outweigh the associated risks. Many reported cases were self-limiting and had a good prognosis with available treatments. Future studies should further evaluate the risk factors, both ocular and systemic, that may contribute to the development of optic neuropathy following COVID-19 vaccination within larger and more diverse populations and elucidate the mechanisms that underly the development of these conditions. This could further assist in the causality assessment of optic neuropathy as an adverse event following COVID-19 vaccination and help optimize the follow-up and treatment of this rare but sight-threatening complication [87].

## Data Availability

Not applicable.

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
