# Peer review of "COVID-19 Vaccine-Associated Optic Neuropathy: A Systematic Review of 45 Patients"

_vaccines, 2022, doi:10.3390/vaccines10101758_

Round 1

Reviewer 1 Report

The manuscript of Prof Elnahry et al consists of the review of a few dozen cases of ADR affecting the eyes in patients exposed to COVID-19 vaccines.

As with any new drug or vaccine (especially if used by millions of individuals), knowledge of the safety profile progressively increases over time, also thanks to such studies.

I have some objections:

1) The article is already hard enough to follow for all the clinical parameters described (they are already in table 1 could be significantly reduced), so referring to the numbers of eyes involved in the suspected adverse reactions complicates even more. I recommend mentioning the number of patients involved and, if anything, how many of them have suffered bilateral damage.

2) For the cases described it can only be stated that there is a correlation of a temporal type, it would have been appropriate to apply to the few cases the assessment of the causality according to the WHO: https://www.who.int/publications/i/item / 9789241516990.

3) The limits are recognized but it should also be emphasized that to date we have a few dozen cases compared to hundreds of millions of subjects exposed to vaccines.

4) It should also be stated that based on the data available today, the frequency of these suspected ADRs is not known and cannot be determined based on available data.

Author Response

Reviewer 1

Thank you for your valuable comments. You will find the responses below in red.

The manuscript of Prof Elnahry et al consists of the review of a few dozen cases of ADR affecting the eyes in patients exposed to COVID-19 vaccines.

As with any new drug or vaccine (especially if used by millions of individuals), knowledge of the safety profile progressively increases over time, also thanks to such studies.

I have some objections:

1) The article is already hard enough to follow for all the clinical parameters described (they are already in table 1 could be significantly reduced), so referring to the numbers of eyes involved in the suspected adverse reactions complicates even more. I recommend mentioning the number of patients involved and, if anything, how many of them have suffered bilateral damage.

We reduced table 1 as the reviewer suggested. We mentioned the number of patients involved where it was missing (lines 119-122).

2) For the cases described it can only be stated that there is a correlation of a temporal type, it would have been appropriate to apply to the few cases the assessment of the causality according to the WHO: https://www.who.int/publications/i/item / 9789241516990.

It is out of the scope and capability of this review to perform a causality assessment of optic neuropathy as an adverse event following COVID-19 vaccination, however, we believe it to be an important step towards doing so by raising more awareness around the topic. Therefore, we have recommended performing such assessment in the future if and when sufficient data is available with reference to the WHO user manual as suggested by the reviewer (lines 249-252).

3) The limits are recognized but it should also be emphasized that to date we have a few dozen cases compared to hundreds of millions of subjects exposed to vaccines.

We have added this limitation as suggested (lines 233-238)

4) It should also be stated that based on the data available today, the frequency of these suspected ADRs is not known and cannot be determined based on available data.

We have added this statement (lines 236-238)

We thank the reviewer for his valuable comments that assisted in improving our manuscript.

Reviewer 2 Report

Major comments:

·         In general, the authors are invited to move their manuscript from a superficial analysis to a deeper and more comprehensive analysis of their results that will benefit readers and open new horizons for complementary studies.

·         Authors request to add a tiny panorama about the link between COVID-19 infection and optic neuropathy (harness this recent report PMID: 34937266), then put your insight about both squalene’s mechanisms from infection and vaccination, which would increase your report's interest.

·         Page 24, lines 168-172, contains unexpected discussion for your analysis through which females represent a majority versus males. Both studies you cited (UK and lee reports) seem to target optic neuropathy in general and the Lee study focused on AION. Is it due to autoimmune disorders and/or others? Please give your deep/comprehensive insight into this variation between females/males. Please look in death into your discussion on page 25, lines 203-218, which contains that “neutralizing antibodies …. “this is an autoimmune reaction due to molecules mimicry, which is already implicated in the induction of autoimmune reactions in covid-19 and post covid-19 and post-vaccination. are just examples for more factors that may be behind your final results analysis (PMID: 35313470, PMID: 31920953, PMID: 32853896, PMID: 35273416, PMID: 35135920)

·         Please let us know the starting and end date for your analysis.

·         Are all COVID-19 vaccines inducing optic neuropathy? this is a very important dimension to include in your text.

 Minor comments:

·         they are requested to mention, at least in the introduction or methods, which optic neuropathy is rare and/or common and does not depend just on “inflammation expression”

Author Response

We thank the reviewer for his valuable comments. He will find our responses below in red.

Major comments:

  • In general, the authors are invited to move their manuscript from a superficial analysis to a deeper and more comprehensive analysis of their results that will benefit readers and open new horizons for complementary studies.

We thank the reviewer for advising us to perform a deeper and more comprehensive analysis.

  • Authors request to add a tiny panorama about the link between COVID-19 infection and optic neuropathy (harness this recent report PMID: 34937266), then put your insight about both squalene’s mechanisms from infection and vaccination, which would increase your report's interest.

We added information from this interesting and relevant article as suggested (lines 214-216).

  • Page 24, lines 168-172, contains unexpected discussion for your analysis through which females represent a majority versus males. Both studies you cited (UK and lee reports) seem to target optic neuropathy in general and the Lee study focused on AION. Is it due to autoimmune disorders and/or others? Please give your deep/comprehensive insight into this variation between females/males.

This was an epidemiologic finding in our study which was supported by previous studies on optic neuropathy. One general explanation may be due to the fact that females in general are known to more likely develop autoimmune diseases especially those that have associated optic neuritis such as MS, while males are more like to develop cardiovascular diseases which are linked to AION. These could be due to hormonal, genetic, and social factors. We have added this in line 180.

Please look in death into your discussion on page 25, lines 203-218, which contains that “neutralizing antibodies …. “this is an autoimmune reaction due to molecules mimicry, which is already implicated in the induction of autoimmune reactions in covid-19 and post covid-19 and post-vaccination. are just examples for more factors that may be behind your final results analysis (PMID: 35313470, PMID: 31920953, PMID: 32853896, PMID: 35273416, PMID: 35135920)

We have enriched our discussion with these references as suggested (lines 214-216).

  • Please let us know the starting and end date for your analysis.

Analysis was done up to September 10, 2022, there was no restriction to start date (line 60).

  • Are all COVID-19 vaccines inducing optic neuropathy? this is a very important dimension to include in your text.

We added this important information (lines 137-140).

 Minor comments:

  • they are requested to mention, at least in the introduction or methods, which optic neuropathy is rare and/or common and does not depend just on “inflammation expression”

This is mentioned in lines 228-229 and the pathophysiology has been emphasized.

We thank the reviewer for his valuable comments.

Reviewer 3 Report

The authors tried to monitor COVID-19 vaccine-associated optic neuropathy within 45 Patients. Overall the manuscript is poorly written. In addition, the number of samples/cases are not enough to have such conclusion. More surveillance studies are required within large population with more diversity of different ages. Time range as well is more important especially with emergence of new variants and efficacy of the used vaccines against them. Also, the parameter used for statistical analyses in this review in not enough and confused.

Author Response

The authors tried to monitor COVID-19 vaccine-associated optic neuropathy within 45 Patients. Overall the manuscript is poorly written. In addition, the number of samples/cases are not enough to have such conclusion. More surveillance studies are required within large population with more diversity of different ages. Time range as well is more important especially with emergence of new variants and efficacy of the used vaccines against them. Also, the parameter used for statistical analyses in this review in not enough and confused.

Thank you for your valuable comments. We have added them in the limitations section and the conclusion has been edited to reflect those limitations.

Round 2

Reviewer 1 Report

The manuscript has improved and I have no further objections

Reviewer 2 Report

Thank you,

Reviewer 3 Report

Overall the manuscript still require extensive English editing.